

# Experimental observation of transient $\delta^{18}$O interaction between snow and
advective airflow under various temperature gradient conditions
Pirmin Philipp Ebner[1], Hans Christian Steen-Larsen[2, 3], Barbara Stenni[4,5], Martin
Schneebeli[1,*], and Aldo Steinfeld[6]
[1] *WSL Institute for Snow and Avalanche Research SLF, 7260 Davos Dorf, Switzerland*
[2] *LSCE Laboratoire des Sciences du Climat et de l'Environnement, Gif-Sur-Yvette Cedex, France*
[3] *Center for Ice and Climate, Niels Bohr Institute, University of Copenhagen, Copenhagen, Denmark*
[4] *Department of Environmental Sciences, Informatics and Statistics, University Ca' Foscari of Venice,*
*Venice, Italy*
[5] *Institute for the Dynamics of Environmental Processes, CNR, Venice, Italy*
[6] *Department of Mechanical and Process Engineering, ETH Zurich, 8092 Zurich, Switzerland*
**Abstract**
Stable water isotopes ($\delta^{18}$O) obtained from snow and ice samples of polar regions
are used to reconstruct past climate variability, but heat and mass transport processes
can affect the isotopic composition. Here we present an experimental study on the effect
on the snow isotopic composition by airflow through a snow pack in controlled
laboratory conditions. The influence of isothermal and controlled temperature gradient
conditions on the $\delta^{18}$O content in the snow and interstitial water vapor is elucidated. The
observed disequilibrium between snow and vapor isotopes led to exchange of isotopes
between snow and vapor under non-equilibrium processes, significantly changing the
$\delta^{18}$O content of the snow. The type of metamorphism of the snow had a significant
influence on this process. These findings are pertinent to the interpretation of the

---

* Corresponding author. email: schneebeli@slf.ch





records of stable isotopes of water from ice cores. These laboratory measurements
suggest that a highly resolved history is relevant for the interpretation of the snow
isotopic composition in the field.
*Keywords*: snow, isotope, isothermal, metamorphism, advection, tomography, post-depositional process
**1. Introduction**

Water stable isotopes in polar snow and ice have been used for several decades as

proxies for global and local temperatures (e.g. Dansgaard, 1964; Lorius et al., 1979;
Grootes et al., 1994; Petit et al., 1999; Johnsen et al., 2001; EPICA Members, 2004).
However, the processes that influence the isotopic composition in high-latitude
precipitation are complex, making direct inference of paleo temperatures from the
isotopic record difficult (Cuffey et al., 1994; Jouzel et al., 1997, 2003; Hendricks et al.,
2000). Several factors affect the vapor and snow isotopic composition, which give rise
to ice core isotopic composition, starting from the process of evaporation in the source
region, until the air mass arrives on top of the ice sheets, and even after snow deposition
(Craig and Gordon, 1964; Merlivat and Jouzel, 1979; Johnsen et al. 2001; Ciais and
Jouzel, 1994; Jouzel and Merlivat, 1984; Jouzel et al., 2003; Helsen et al., 2005, 2006,
2007; Cuffey and Steig, 1998; Krinner and Werner, 2003). Mechanical processes such
as mixing, seasonal scouring, or spatial redistribution of snow can alter seasonal and
annual records (Fisher et al., 1983; Hoshina et al., 2014). Post-depositional processes
associated with wind scouring and snow redistribution are known to introduce a "post-
depositional noise" in the surface snow. Comparisons of isotopic records obtained from



nearby shallow ice cores have allowed for estimation of a signal-to-noise ratio and a
common climate signal (Fisher and Koerner, 1988, 1994; White et al., 1997; Steen-
Larsen et al., 2011; Sjolte et al., 2011; Masson-Delmotte et al., 2015). After deposition,
interstitial diffusion in the firn and ice affects the water-isotopic signal but back-
diffusion or deconvolution techniques have been used to establish the original isotope
signal (Johnsen, 1977; Johnsen et al., 2000).

The interpretation of ice core data and the comparison with atmospheric model

results implicitly rely on the assumption that the snowfall precipitation signal is
preserved in the snow-ice matrix (Werner et al. 2011). Classically, ice-core stable-
isotope records are interpreted as reflecting precipitation-weighted signals, and
compared to observations and atmospheric model results for precipitation, ignoring
snow-vapor exchanges between surface snow and atmospheric water vapor (e.g. Persson
et al., 2011). However, recent studies carried out on top of the Greenland and Antarctic
ice sheets combining continuous atmospheric water vapor isotope observations with
daily snow surface sampling document a clear day-to-day variation of isotopic
composition of surface snow in-between precipitation events as well as diurnal change
in the snow isotopes (Steen-Larsen et al., 2014a; Ritter et al., 2016, Casado et al., 2016).
This effect was interpreted as being caused by the uptake of the synoptic-driven
atmospheric water-vapor isotope signal by individual snow crystals undergoing snow
metamorphism (Steen-Larsen et al., 2014a) and the diurnal variation in moisture flux
(Ritter et al., 2016). However, the impact of this process on the isotope-temperature
reconstruction is not yet sufficiently understood, but crucial to constrain. This process,
compared to interstitial diffusion (Johnsen, 1977; Johnsen et al., 2000), will alter the
isotope mean value. The field observations challenge the previous assumption that
sublimation occurred molecular layer-by-layer with no resulting isotopic fractionation




(Dansgaard, 1964; Friedman et al., 1991; Town et al., 2008; Neumann and Waddington,
2004). It is assumed that the solid undergoing sublimation would not be unduly enriched
in the heavier isotope species due to the preferential loss of lighter isotopic species to
the vapor (Dansgaard, 1964; Friedman et al., 1991). Because self-diffusion in the ice is
about three orders of magnitude slower than molecular diffusion in the vapor, the
amount of isotopic separation in snow is assumed to be negligible.

Snow is a bi-continuous material consisting of fully connected ice crystals and pore

space (air) (Löwe et al., 2011). Because of the proximity to the melting point, the high
vapor pressure causes a continuous recrystallization of the snow microstructure known
as snow metamorphism, even under moderate temperature gradients (Pinzer et al.,
2012). The whole ice matrix is continuously recrystallizing by sublimation and
deposition, with vapor diffusion as the dominant transport process. Pinzer et al. (2012)
showed that a typical half-life of the ice matrix is a few days. The intensity of the
recrystallization is dictated by the temperature gradient. Temperature, and geometrical
factors (porosity and specific surface area) also play a significant role (Pinzer and
Schneebeli, 2009; Pinzer et al., 2012). Snow has a high permeability (Calonne et al.,
2012; Zermatten et al., 2014), which facilitates diffusion of gases and, under appropriate
conditions, airflow (Gjessing, 1977; Colbeck, 1989; Sturm and Johnson, 1991;
Waddington et al., 1996). Modeling of the influence of the so-called 'wind pumping'-
effect (Fisher et al., 1983; Neumann and Waddington, 2004), in which the interstitial
water vapor is replaced by atmospheric air pushed through the upper meters of the snow
pack by small scale high and low pressure areas caused by irregular grooves or ridges
formed on the snow surface (dunes and sastrugi), have assumed that the snow grains
would equilibrate with the interstitial water vapor on timescales governed by ice self-
diffusion. However, no experimental data are available to support this assumption.



With this in mind the experimental study presented here is specifically developed to
investigate the effect of ventilation inside the snow pack on the isotopic composition.
Only conditions deeper than 1 cm inside a snowpack are considered. Previous work
showed that (1) under isothermal conditions, the Kelvin effect leads to a saturation of
the pore space in the snow but does not affect the structural change (Ebner et al.,
2015a); (2) applying a negative temperature gradient along the flow direction leads to a
change in the microstructure due to deposition of water molecules on the ice matrix
(Ebner et al., 2015b); and (3) a positive temperature gradient along the flow had a
negligible total mass change of the ice but a strong reposition effect of water molecules
on the ice grains (Ebner et al., 2016). Here, we measured continuously the isotopic
composition of an airflow containing water vapor through a snow sample under both
isothermal and temperature gradient conditions. Micro computed-tomography (μCT)
was applied to obtain the 3D microstructure and morphological properties of snow.
**2. Experimental setup**
Isothermal and temperature gradient experiments with fully saturated airflow and
defined isotopic composition were performed in a cold laboratory at around $T_{lab} \approx$ -15
°C with small fluctuations of ± 0.8 °C (Ebner et al., 2014). Snow produced from de-
ionized tap water in a cold laboratory (water temperature: 30 ºC; air temperature: -20
ºC) was used for the snow sample preparation (Schleef et al., 2014). The snow was
sieved with a mesh size of 1.4 mm into a box, and isothermally sintered for 27 days at -
5 ºC to increase the strength, in order to prevent destruction of the snow sample due to
the airflow, and to evaluate the effect of metamorphism of snow. The morphological
properties of the snow are listed in Table 1. The sample holder (diameter 53 mm, height
30 mm, 0.066 liter) was filled by a cylinder cut out from the sintered snow. To prevent
that air can flow between the snow sample and the sample holder walls, the undisturbed



snow disk was filled in at a higher temperature (about -5 °C) and sintering was allowed
for about 1 h before cooling down and start of the experiment. The setup of Ebner et al.
(2014) was modified by additionally inserting a water vapor isotope analyzer (Model:
L1102-I Picarro, Inc., Santa Clara, CA, USA) to measure the isotopic ratio $\delta^{18}O$ of the
water vapor contained in the airflow at the inlet and outlet of the sample holder. The
experimental setup consisted of three main components (humidifier, sample holder, and
the Picarro analyzer) were connected with insulated copper tubing and Swagelok fitting
(Fig. 1). The tubes to the Picarro analyzer were heated to prevent deposition of water
vapor and thereby fractionation. The temperature was monitored with thermistors inside
the humidifier and at the inlet and outlet of the snow sample. A dry air pressure tank
controlled by a mass flow controller (EL-Flow, Bronkhorst) generated the airflow. A
humidifier, consisting of a tube (diameter 60 mm, height 150 mm, 0.424 liter volume)
filled with crushed ice particles (snow from Antarctica with low $\delta^{18}O$ composition), was
used to saturate the dry air entering the humidifier with water vapor at an almost
constant isotopic composition. The air temperature in the humidifier and at the inlet of
the snow sample was maintained at the same value (accuracy ± 0.2 K) to limit changes
influence of variability in absolute vapor pressure and isotopic composition. We
measured the $\delta^{18}O$ of the water vapor produced by the humidifier before and after each
experimental run ($\delta^{18}O_{hum}$). The outlet flow ($\delta^{18}O_a$) of the sample holder was
continuously measured during the experiment to analyze the temporal evolution of the
isotopic signal. All data from the Picarro analyzer were corrected to the humidity
reference level using the established instrument humidity-isotope response (2013;
2014b). In addition, VSMOW-SLAP correction and drift correction were performed.
We followed the calibration protocol and used the calibration system described in detail
by Steen-Larsen et al. (2013; 2014b).



The sample holder described by Ebner et al. (2014) was used because it already had
the appliance to analyze the snow by μCT. Tomography measurements were performed
with a modified μ-CT80 (Scanco Medical). The equipment incorporated a microfocus
X-ray source, operated at 70 kV acceleration voltage with a nominal resolution of 18
μm. The samples were scanned with 1000 projections per 180º in high-resolution
setting, with typical adjustable integration time of 200 ms per projection. The field of
view of the scan area was 36.9 mm of the total 53 mm diameter, and subsamples with a
dimension of $7.2 \times 7.2 \times 7.2$ mm$^3$ were extracted for further processing. The
reconstructed μCT images were filtered using a $3 \times 3 \times 3$ median filter followed by a
Gaussian filter ($\sigma = 1.4$, support = 3). The Otsu method (Otsu, 1979) was used to
automatically perform clustering-based image thresholding to segment the grey-level
images into ice and void phase. Morphological properties in the two-phase system were
determined based on the exact geometry obtained by the μCT. Tetrahedrons
corresponding to the enclosed volume of the triangulated ice matrix surface were
applied on the segmented data to determine porosity ($\varepsilon$) and specific surface area (SSA).
Opening size distribution operation was applied in the segmented μCT data to extract
the mean pore size ($d_{mean}$) (Haussener et al., 2012).
Three experiments with saturated advective airflow through the snow sample were
performed to record the following parameters and analyze their effects: (1) isothermal
conditions to analyze the influence of curvature effects (Kaempfer al et., 2007); (2)
positive temperature gradient applied to the snow sample where cold air is heated up
while flowing through the sample in order to analyze the influence of sublimation; (3)
negative temperature gradient applied to the snow sample where warm air is cooled
down while flowing across the sample, to analyze the influence of net deposition.
During the temperature gradient experiments, a temperature difference of 1.4 ºC and 1.8



ºC was imposed resulting in a gradient of +47 K m$^{-1}$ and -60 K m$^{-1}$, respectively. The
runs were performed at atmospheric pressure and with a volume flow rate of 3.0 liter
min$^{-1}$ corresponding to an average flow speed in the pores of ≈ 30 mm s$^{-1}$. In experiment
(2) the outlet temperature and in experiment (3) the inlet and also the humidifier
temperature was actively controlled using thermo-electric elements. Variations in
temperature of up to ± 0.8 ºC were due to temperature fluctuations inside the cold
laboratory, leading to slightly variable temperature gradients and mean temperature in
experiment (2) and (3). Table 1 presents a summary of the experimental conditions and
the morphological properties of the snow samples. At the end of each experiment, the
snow sample was cut into five layers of 6 mm height and the isotopic composition of
each layer was analyzed to examine the spatial $\delta^{18}O$ gradient in the isotopic composition
of the snow sample.
A slight increase with a maximum of 0.7 ‰ of $\delta^{18}O$ in the water vapor produced by
the humidifier was observed during the experiments (Table 2). This change of ~0.7‰ is
not significant compared to the difference between the isotopic composition of the water
vapor and the snow sample in the sample holder of ~53‰ and the temporal change of
the water vapor isotopes on the back side of the snow sample.

**3. Results**

**3.1 Isothermal condition**

The experiment (1) was performed for 24 h at a mean temperature of $T_{mean}$ = -15.5
ºC. $\delta^{18}O_a$ decreased exponentially in the outlet flow was observed throughout the
experimental run as shown in Fig. 2. Initially, the $\delta^{18}O_a$ content in the flow was -27.7 ‰
and exponentially decreased to -47.6 ‰ after 24 h. The increase of $\delta^{18}O_a$ in the first
approximately 30 min was due to memory effects from air and possible condensed





water left in the tubes from a prior experiment. The small fluctuations in the $\delta^{18}O_a$
signal at $t \approx 7$ h, 17 h and 23 h were due to small temperature changes in the cold
laboratory.
We observed a strong interaction between the airflow and the snow was observed in
the isotopic composition of the snow. The $\delta^{18}O_s$ signal in the snow decreased by 4.75 –
7.78 ‰ and an isotopic gradient in the snow was observed after the experimental run,
shown in Fig. 3. Initially, the snow had a homogeneous isotopic composition of $\delta^{18}O_s =$
-10.97 ‰ but post-experiment sampling showed a decrease in the snow $\delta^{18}O$ at the inlet
side to -17.75 ‰ and at the outlet side to -15.72 ‰. The spatial $\delta^{18}O_s$ gradient of the
snow had an approximate slope of 0.68 ‰ $mm^{-1}$ at the end of the experimental run.
Table 2 shows the $\delta^{18}O$ value in snow at the beginning ($t = 0$) and end ($t = 24$ h) of the
experiment.

### 209    3.2 Air warming by a positive temperature gradient along the airflow

The experiment (2) was performed over a period of 24 h with an average
temperature gradient of approximatively +47 K $m^{-1}$ (warmer temperatures at the outlet
of the snow) and an average mean temperature of -14.7 ºC. We observed again a
relaxing exponential decrease of $\delta^{18}O_a$ in the outlet flow was observed throughout the
measurement period as shown in Fig. 2, but the decrease was slower compared to the
isothermal run. Initially, the $\delta^{18}O_a$ content in the flow coming through the snow disk
was -29.8 ‰ and exponentially decreased to -41.9 ‰ after 24 h. The increase of $\delta^{18}O_a$
in the first 30 min was due to memory effects as explained previously. The small
fluctuations in the $\delta^{18}O_a$ signal at $t \approx 2.7$ h, and 12.7 h were due to small temperature
changes in the cold laboratory.
The $\delta^{18}O_s$ signal in the snow decreased up to 4.66 – 7.66 ‰ and a gradient in
isotopic composition of the snow was observed after the experimental run, shown in



Fig. 3. Initially, the snow had a homogeneous isotopic composition of $\delta^{18}O_s$ = -11.94
‰, but post-experiment sampling showed a decrease at the inlet side to -19.6 ‰ and at
the outlet side to -16.6 ‰. The spatial $\delta^{18}O_s$ gradient of the snow had an approximate
slope of 1.0 ‰ mm$^{-1}$ at the end of the experimental run. Table 2 shows the $\delta^{18}O_s$ value
in snow at the beginning ($t = 0$) and end ($t = 24$ h) of the experiment.
**3.3 Air cooling by a negative temperature gradient along the air flow**
The experiment (3) was performed for 84 h instead of 24 h to better estimate the
trend in $\delta^{18}O_a$ in the outlet flow. An average temperature gradient of approximately -60
K m$^{-1}$ (colder temperatures at the outlet of the snow) and an average mean temperature
of -13.2 ºC were observed during the experiment. As in the previous experiments, a
relaxing exponential decrease of $\delta^{18}O_a$ in the outlet flow was observed throughout the
experimental run as shown in Fig. 2, but the decrease was slower compared to the
isothermal run and temperature gradient opposed to the airflow. Initially, the $\delta^{18}O_a$
content in the flow was -29.8 ‰ and exponentially decreased to -37.7 ‰ after 84 h. The
increase of $\delta^{18}O_a$ in the first 30 min was again due to memory effects. The small
fluctuations in the $\delta^{18}O_a$ signal at $t \approx 7.3$ h, 21.3 h, 31.3 h, 45.3 h, 55.3 h, 69.3 h, and
79.3 h were due to small temperature changes in the cold laboratory.
The $\delta^{18}O_s$ signal in the snow decreased up to 4.46 – 15.09 ‰ and a gradient in the
isotopic composition of the snow was observed after the experimental run, shown in
Fig. 3. Initially, the snow had an isotopic composition of $\delta^{18}O_s$ = -10.44 ‰ but post-
experiment sampling showed a decrease at the inlet side to -25.53 ‰ and at the outlet
side to -15.00 ‰. The spatial $\delta^{18}O_s$ gradient of the snow had an approximate slope of
3.5 ‰ mm$^{-1}$ at the end of the experimental run. Table 2 shows the $\delta^{18}O_s$ value in snow
at the beginning ($t = 0$) and end ($t = 84$ h) of the experiment.

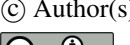



## 4. Discussion


All experiments showed a strong exchange in $\delta^{18}$O between the snow and water-
vapor saturated air resulting in a significant change of the value of the stable isotopes in
the snow. The advective conditions in the experiments were comparable with surface
snow layers in Antarctica and Greenland, but at higher temperature, especially
compared to interior Antarctica. In wind pumping theory, an airflow velocity of $u_D \approx 10$
mm s$^{-1}$ (corresponding Reynolds number Re $\approx$ 0.65) was estimated inside the surface
snow layers ($d_{mean} \approx 1$ mm) for a high wind speed above the snow surface ($\approx 10$ m s$^{-1}$)
(Neumann, 2003). We performed experiments with airflow velocities inside the snow
sample of around 30 mm s$^{-1}$ (corresponding Reynolds number Re = 0.7), which was a
factor three higher than in natural conditions, but still in the feasible flow regime
according to the Reynolds number.
The results also showed strong interactions in $\delta^{18}$O between snow and air depending
on the different temperature gradient conditions. The experiments indicate that
temperature variation and airflow above and through the snow structures (Sturm and
Johnson, 1991; Colbeck, 1989; Albert and Hardy, 1995) seem to be dominant processes
affecting water stable isotopes of surface snow. In a typical Antarctic and Greenland
snow profile, strong interactions between the atmosphere and snow occurs, especially in
the first 2 m (Neumann and Waddington, 2004; Town et al., 2008), called the
convective zone. In the convective zone, air can move relatively freely and therefore
exchange between snow and the atmospheric air occurs. Air flowing into the snow
reaches saturation vapor pressure nearly instantly through sublimation (Neumann et al.,
2008; Ebner et al., 2015a). Our results support the interpretation that changes in surface
snow isotopic composition are expected to be significant if large day-to-day surface
changes in water vapor occur in between precipitation events, wind pumping is efficient





and snow metamorphism is enhanced by temperature gradients in the upper first
centimeters of the snow (Steen-Larsen et al., 2014a).

We expect that our findings will lead to influence the interpretation of the water

stable isotope records from ice cores. Classically, ice core stable isotope records are
interpreted as paleo-temperature reflecting precipitation-weighted signals. When
comparing observations and atmospheric model results for precipitation with ice core
records, such vapor-snow exchanges are normally ignored (e.g. Persson et al. (2011)
and Fujita and Abe (2006)). However, vapor-snow exchange enhanced by
recrystallization rate seems to be an important factor for the high variation in the snow
surface $\delta^{18}O$ signal as supported by our experiments. It was hypothesized that the
changes in the snow-surface $\delta^{18}O$ reported by Steen-Larsen et al. (2014a) are caused by
changes in large-scale wind and moisture advection of the atmospheric water vapor
signal and snow metamorphism. The strong interaction between atmosphere and near-
surface snow can modify the ice core water stable isotope records.

The rate-limiting step for isotopic exchange in the snow is isotopic equilibration

between the pore-space vapor and surrounding ice grains. The relaxing exponential
decrease of $\delta^{18}O$ in the outflow of our experiments predicted that full isotopic
equilibrium between snow and atmospheric vapor will not be reached at any depth
(Waddington et al., 2002; Neumann and Waddington, 2004) but changes towards
equilibrium with the atmospheric state occurs, effectively changing the "target" towards
which the snow is equilibrating (Steen-Larsen et al., 2013, 2014a).

As snow accumulates, the upper 2 m are advected through the ventilated zone

(Neumann and Waddington, 2004; Town et al., 2008). In areas with high accumulation
rate (e.g. South Greenland), snow is advected for a short time through the ventilated
zone. Relatively short time vapor snow exchange would result in higher spatial



variability compared to long-time. However, the effects of snow ventilation on isotopic
composition may become more important as the accumulation rate of the snow
decreases (< 50 mm a$^{-1}$), such that snow remains in the near-surface ventilated zone for
many years (Waddington et al., 2002; Hoshina et al., 2014; Hoshina et al., 2016). As the
snow remains longer in the near-surface ventilated zone, a larger $\delta^{18}$O exchange
between snow and atmospheric vapor will occur. Consequently, the isotopic content of
layers at sites with high and low accumulation rates can evolve differently, even if the
initial snow composition had been equal, and the sites had been subjected to the same
histories of air-mass vapor.

We now discuss the fact that despite a relatively small change in the difference

between the isotopic composition of the incoming vapor and the snow, large differences
in the isotopic composition of the water vapor at the outlet flow exist for the three
different experimental setups. Based on the difference in the outlet water vapor isotopic
composition, we hypothesized that different processes are at play for the different
experiments.  It is obvious that there is a fast isotopic exchange with the surface of the
ice crystals, and a much slower timescale on which the interior of the ice crystals gets
altered. Due to the low diffusivity of H$_2^{16}$O and H$_2^{18}$O in ice ( $D_{H_2^{18}O} \approx D_{H_2^{16}O} = \sim 10^{-15}$
m$^2$ s$^{-1}$ (Ramseier, 1967; Johnsen et al., 2000)), we assumed that the interior is not
altered on the timescale of the experiment. This explained why the net isotopic change
of the bulk sample is relatively small compared to the changes in the outlet water vapor
isotopes. The effective 'ice-diffusion depth' of the isotopic exchange during the
experiments is given as $L_D = \sqrt{D \cdot t}$ , where $D$ is the diffusion coefficient of H$_2^{16}$O and
H$_2^{18}$O in ice, respectively, and $t$ the experimental time. The calculated 'ice-diffusion
depth' $L_D$, is ~ 9.3 μm for experiment (1) and (2), and ~17.4 μm for experiment (3),
respectively, indicating an expected low altering of the interior of the ice crystal.





However, snow has a large specific surface area and therefore a high exchange area.
This has an effect on the $\delta^{18}O$ snow concentration. The fraction of the total volume $V_{tot}$
of ice that is close enough to the ice surface to be affected by diffusion in time $t$ is then
$\rho_{ice} \cdot SSA \cdot L_D$, where SSA is the specific surface area (area per unit mass), and $L_D$ is
the diffusion depth (above) for time $t$. For $t \approx 24$ hours, a large fraction (24 to 43 %) the
total volume $V_{tot}$ of the ice matrix can be accessed through diffusion. It is quite hard to
see the total $\delta^{18}O$ snow difference between experiments (1) and (2) after the experiment
compared to the $\delta^{18}O$ of the vapor in the air at the outlet. But there is still a notable
effect in the $\delta^{18}O$ of the snow between experiment (1) and (2). Due to the higher
recrystallization rate of experiment (2) the spatial $\delta^{18}O_s$ gradient of the snow (1.0 ‰
$mm^{-1}$) is higher than for experiment (1) (0.68 ‰ $mm^{-1}$). Increasing the experimental
time, the $\delta^{18}O$ change in the snow increases (experiment (3)). In general, the calculated
'ice-diffusion depth' is realistic under isothermal conditions where diffusion processes
are the main factors (Kaempfer and Schneebeli, 2007; Ebner et al., 2015). Applying a
temperature gradient, the impact of diffusion is suppressed due to the high
recrystallization rate by sublimation and deposition. Due to the low half-life of the ice
matrix of a few days, the growth rates are typically on the order of 100 μm per day
(Pinzer et al., 2012). Therefore, this redistribution of ice counteracts the diffusion into
the solid ice.

By comparing similarities and differences between the outcomes of the three

experimental setups we will now discuss the physical processes influencing the
interaction and exchange processes within the snowpack between the snow and the
advected vapor. We first notice that the final snow isotopic profile of experiment (1)
(isothermal) and (2) (positive temperature gradient along the direction of the flow) are
comparable to each other. Despite this similarity, the evolution in the outlet water vapor





of experiment (1) showed a significantly stronger depletion compared to experiment (2).
For experiment (3) (negative temperature gradient along the direction of the flow) we
observed the smallest change in outlet water vapor isotopes but the largest snow-pack
isotope gradient after the experiment. However, this change was caused by 84 hours
flow instead of 24 hours.
Curvature effects, temperature gradients and therefore the recrystallization rate
influence the mass transfer of $H_2^{16}O$/ $H_2^{18}O$ molecules. The higher the recrystallization
rate of the snow the slower the adaption of the outlet air concentration to the inlet air
concentration (see in experiment (2) and (3)). Under isothermal conditions (experiment
(1)) the only effect influencing the recrystallization rate is the curvature effect
(Kaempfer and Schneebeli, 2007). However, based on the experimental observations
(Kaempfer and Schneebeli, 2007) this effect decreases with decreasing temperature and
increasing experimental time. Applying an additional temperature gradient on a snow
sample, there are complex interplays between local sublimation and deposition on
surfaces and the interaction of water molecules in the air with the ice matrix due to
changing saturation conditions of the airflow. Therefore, the recrystallization rate
increases and thus the change in the $\delta^{18}O$ of the air. For experiment (2) there is a
complex interplay between sublimation and deposition of water molecules into the
interstitial flow (Ebner et al., 2015c) while for experiment (3) there are deposition of
molecules carried by the interstitial flow onto the snow crystals (Ebner et al., 2015b).
Furthermore, in the beginning of each experiment there is a tendency to sublimate from
edges of the individual snow crystals due to the higher curvature. As the edges were
sublimated and deposition occurred in the concavities, the individual snow crystals
became more rounded, slowing down the transfer of water molecules into the interstitial
airflow. We noticed for all three experiments that within the uncertainty of the isotopic



composition of the snow, the initial isotopic composition of the vapor was the same and
in isotopic equilibrium with the snow. The difference between experiment (1) and (2)
lies in the fact that due to the temperature gradient in experiment (2) there is an
increased transfer of water vapor with the isotopic composition of the snow in to the
airflow. Hence the depleted air from the humidifier advected through the snow disk is
mixed with a relatively larger vapor flux from the snow crystals. Additionally, we also
expected less deposition into the concavities in experiment (2) compared to experiment
(1). However, it is interesting to note that the final isotopic profile of the snow disk is
similar in experiment (1) and experiment (2). We interpreted this as being a result of
two processes acting in opposite direction: although relatively isotope-depleted vapor
from the humidifier was deposited on the ice matrix there was also a higher amount of
sublimation of relatively isotope-enriched vapor from the snow disk in experiment (2).
Experiment (3) separates itself from the other two experiments in the way that as the
water vapor from the humidifier is advected through the snow disk there is a continuous
deposition of very depleted air due the negative temperature gradient. As for the case of
experiment (1) and (2) there was also in experiment (3) a constant sublimation of the
convexities into the vapor stream. We notice that despite the fact that experiment 3 ran
for 84 hours the snow at the outlet side of the snow-disk did not become more
isotopically depleted compared to experiment (1) and (2). However, the snow on the
inlet side became significantly more isotopically depleted. By having this observation in
mind, together with the fact that the vapor of the outlet of the snow-disk is less depleted
compared to experiment (1) and (2), lead us to hypothesize that there is a relatively
larger deposition of isotopically depleted vapor from the humidifier as the vapor is
advected through the snow disk. This means that a relatively larger component of the





isotopic composition of the vapor is originating by sublimation from the convexities of
the snow disk and less from the isotopically depleted vapor from the humidifier.

Our hypotheses ask for additional validation by more detailed experiments.

Specifically, it would be crucial to know the mass balance of the snow disk more
precisely, which could be done by scanning the entire snow disk following the change
in density and morphological properties over the entire height, or gravimetrically.
Insights would also be achieved with experiments using snow of the same isotopic
composition, but different SSA, as this would allow calculating more precisely the
different observed exchange rates. Additionally, different and colder background
temperatures should be tested to better understand inland Antarctic environment and the
effect of the quasi-liquid layer, which is necessary for the development of a numerical
model. Isotopically different combinations of vapor and snow should be performed. In
the present manuscript, vapor with low $\delta^{18}O$ isotopic composition was transported
through snow with relative high $\delta^{18}O$ isotopic composition. It would be interesting to
reverse the combination and perform experiments with different combinations to
provide more insights on mass and isotope exchanges between vapor and snow.
Experiments with longer running time helps to understand the change in the ice matrix
better. Further, as in the humidifier we had not only sublimation but a complex interplay
of simultaneously sublimation and deposition due to the geometrical complexity of
snow, a further experiment is suggested to show isotopic fractionation during
sublimation. Dry air is blown over a flat ice surface and is immediately removed. With
the suggested experiment the actual statement whether there is no measurable
fractionation and the sublimated vapor has the same $\delta^{18}O$ value as the sublimating ice or
not can be proved. Based on the results of this paper, we expect a different $\delta^{18}O$ value
between vapor and ice, especially at the beginning (see Fig. 4). Due to the fact that





$H_2^{16}O$ is lighter in mass than $H_2^{18}O$, the energy required to change the state from solid
to vapor is less. Therefore, especially at the beginning of the experiment, the vapor is
depleted in $H_2^{18}O$ and thus a different $\delta^{18}O$ value in the vapor and ice should be seen.
After a while the ice interface is depleted in $H_2^{16}O$ molecules and because the self-
diffusion in ice is low, an approach of the $\delta^{18}O$ value of the vapor and ice should be
seen. Comparing the penetration depths for diffusion ($L_D = \sqrt{D \cdot t}$; where $D \approx 10^{-15}$ m$^2$ s$^-$
$^1$) and for sublimation ($L_{sub} = v_{sub}t$; where $v_{sub} \approx 100$ µm per day (Pinzer et al., 2012))
the time $t = D/v_{sub} \approx 10^3$ s represents the transition from diffusion-dominated behavior
(with fractionation on sublimation) to sublimation-dominated behavior (in which nearly
all $H_2^{18}O$ atoms are forced to enter the vapor phase). This is obviously not a precise
number but it suggests that after about $10^3$ seconds, sublimation from the ice crystals
occurs without significant isotopic fractionation. However, as we have not performed
such experiments yet, these statements are only speculative.
**4. Summary and conclusion**
We analyzed the influence of airflow and metamorphism across a snow sample on
the $\delta^{18}O$ isotopic composition in controlled laboratory experiments. Three experiments
with saturated advective airflow across the snow sample with a volume flow rate of 3.0
liter min$^{-1}$ ($u_D \approx 3$ cm s$^{-1}$) were performed: (1) isothermal run to analyze the influence of
the curvature effects; (2) positive temperature gradient run (approx. +47 K m$^{-1}$) along
the airflow where cold air heated up while flowing through the sample to analyze the
influence of net ice mass loss in a snowpack; and (3) negative temperature gradient run
(approx. -60 K m$^{-1}$) where warm air cooled down while flowing through the sample, to
analyze the influence of deposition. Air with low $\delta^{18}O$ content (-68 ‰ − -62 ‰) was
blown into the snow sample and the $\delta^{18}O$ outlet was continuously measured with a





water vapor isotope analyzer. The $\delta^{18}O$ distribution through the snow disk was
measured at the beginning and end of each experiment. µCT measurements were
applied to obtain the 3D microstructure and the morphological properties, namely:
porosity ($\varepsilon$), specific surface area (SSA), and the mean pore size ($d_{\mathrm{mean}}$) of the snow.

Laboratory experimental runs were performed where a transient $\delta^{18}O$ interaction

between snow and air was observed. The airflow altered the isotopic composition of the
snowpack and supports an improved climatic interpretation of ice core stable water
isotope records. The water vapor saturated airflow with an isotopic difference of up to
55‰ changed within 24 h and 84 h the original $\delta^{18}O$ isotope signal in the snow by up to
7.64 ‰ and 15.06 ‰. The disequilibrium between snow and air isotopes led to the
observed exchange of isotopes, the rate depending on the temperature gradient
conditions. Concluding, increasing the recrystallization rate in the ice matrix the
temporal change of the $\delta^{18}O$ concentration at the outflow decreases (experiment (2) and
(3)). Decreasing the recrystallization rate the temporal curve of the outlet concentration
is getting steeper reaching the $\delta^{18}O$ inlet concentration of the air faster (experiment (1)).

Additionally, the complex interplay of simultaneous diffusion, sublimation and

deposition due to the geometrical complexity of snow has a strong effect on the $\delta^{18}O$
signal in the snow and cannot be neglected. A temporal signal can be superimposed on
that cloud-temperature signal, (a) if the snow remains near the surface for a long time,
i.e. in a low-accumulation area, and (b) is exposed to a history of air masses carrying
vapor with a significantly different isotopic signature than the precipitated snow.

These are novel measurements and will therefore be important for allowing other

researchers formulate their research question based on and carry out further
experiments. Our results represent the first direct experimental observation showing
interaction between the water isotopic composition of the snow, the water vapor in the



air and recrystallization due to temperature gradients. Previous work on isotopic content
of surface snow failed to incorporate the recrystallization process, and recrystallization
and bulk mass exchange must be incorporated into future models of snow and firn
evolution. Further studies are required on the influence of temperature and airflow as
well as snow microstructure on the mass transfer phenomena for validating the
implementation of stable water isotopes in snow models.


### Acknowledgements

The Swiss National Science Foundation granted financial support under project Nr.
iso. H.C. Steen-Larsen was supported by the AXA Research Fund. The authors thank K.
Fujita, E. D. Waddington and an anonymous reviewer for the suggestions and critical
review. M. Jaggi, S. Grimm, A. Schlumpf, and S. Berben gave technical support. The
data for this paper are available by contacting the corresponding author.

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



**Table 1:** Morphological properties and flow characteristics of the experimental runs:
snow density ($\rho$), porosity ($\varepsilon$), specific surface area per unit mass (SSA), mean pore
space diameter ($d_{\text{mean}}$), superficial velocity in snow ($u_{\text{D}}$), corresponding Reynolds
number (Re = $d_{\text{mean}} \cdot u_{\text{D}}/v_{\text{air}}$), average inlet temperature of the humidifier and at the inlet
($T_{\text{in,mean}}$), average outlet temperature at the outlet ($T_{\text{out,mean}}$), and average temperature
gradient ($\nabla T_{\text{ave}}$). Experiment (1) corresponds to the isothermal conditions; Experiment
(2) to air warming; and Experiment (3) to air cooling in the snow sample.

|  | $\rho$ | $\varepsilon$ | SSA | $d_{\text{mean}}$ | $u_{\text{D}}$ | Re | $T_{\text{in,mean}}$ | $T_{\text{out,mean}}$ | $\nabla T_{\text{ave}}$ |
|---|---|---|---|---|---|---|---|---|---|
|  | kg m$^{-3}$ | – | m$^2$ kg$^{-1}$ | mm | m s$^{-1}$ | – | °C | °C | K m$^{-1}$ |
| Experiment (1) | 201.74 | 0.78 | 28.0 | 0.39 | 0.03 | 0.76 | -15.5 | -15.5 | – |
| Experiment (2) | 201.74 | 0.78 | 29.7 | 0.36 | 0.03 | 0.70 | -15.4 | -14.0 | +47 |
| Experiment (3) | 220.08 | 0.76 | 27.2 | 0.37 | 0.031 | 0.74 | -12.3 | -14.1 | -60 |






**Table 2:** $\delta^{18}O$ in the vapor in the humidifier ($\delta^{18}O_{hum}$) and of the snow in the sample
holder ($\delta^{18}O_s$) at the beginning ($t = 0$) and end ($t = $ end) of each experiment and the final
$\delta^{18}O$ content of the snow in the sample holder at the inlet ($z = 0$ mm) and outlet ($z = 30$
mm). Experiment (1) corresponds to the isothermal conditions; Experiment (2) to air
warming; and Experiment (3) to air cooling in the snow sample.

| | $\delta^{18}O_{hum}$ | | $\delta^{18}O_{s,\, t=0}$ | $\delta^{18}O_{s,\, t=\text{end}}$ | |
| --- | --- | --- | --- | --- | --- |
| | ‰ | | ‰ | ‰ | |
| | $t = 0$ | $t = $ end | | $z = 0$ mm | $z = 30$ mm |
| Experiment (1) | -68.2 | -67.5 | -10.97 | -17.75 | -15.72 |
| Experiment (2) | -66.3 | -66.1 | -11.94 | -19.60 | -16.60 |
| Experiment (3) | -62.8 | -62.2 | -10.44 | -25.53 | -15.00 |






**Figure captions**
**Fig. 1.** Schematic of the experimental setup. A thermocouple (TC) and a humidity
sensor (HS) inside the humidifier measured the the mean temperature and
humidity of the airflow. Two thermistors (NTC) close to the snow surface
measured the inlet and outlet temperature of the airflow (Ebner et al., 2014).
The Picarro Analyzer measured the isotopic composition $\delta^{18}O$ of the outlet
flow. Inset: 3D structure of $110 \times 42 \times 110$ voxels ($2 \times 0.75 \times 2$ mm$^3$)
obtained by the μCT.
**Fig. 2.** Temporal isotopic composition of $\delta^{18}O$ of the outflow for each of the
experimental run. The spikes in the $\delta^{18}O$ were due to small temperature
changes in the cold laboratory (Ebner et al., 2014). Exp. (1) corresponds to
the isothermal conditions; Exp. (2) to air warming; and Exp. (3) to air
cooling in the snow sample. The higher the recrystallization rate of the snow
the slower the adaption of $\delta^{18}O$ of the outlet air to the inlet air. The
illustration in the lower right corner shows the relation between $\delta^{18}O$ of the
initial snow, inlet, and outlet of the air.
**Fig. 3.** Spatial isotopic composition of $\delta^{18}O$ of the snow sample at the beginning ($t$
$= 0$) and at the end ($t =$ end) for each experiment. The air entered at $z = 0$
mm and exited at $z = 30$ mm. Exp. (1) corresponds to the isothermal
conditions; Exp. (2) to air warming; and Exp. (3) to air cooling in the snow
sample.
**Fig. 4.** Schematic of isotopic fractionation of vapor and ice during sublimation.
Water molecules sublimate rapidly from a flat ice surface in dry air and are
immediately removed. The different in relative mass of $H_2^{16}O$ and $H_2^{18}O$ led
to an isotopic fractionation of vapor and ice. The isotopic fractionation is a





brief transient effect, possibly lasting only a few minutes. Time $t$ represents
the transition from diffusion-dominated behavior (with fractionation on
sublimation) to sublimation-dominated behavior (in which nearly all $H_2^{18}O$
atoms are forced to enter the vapor phase).

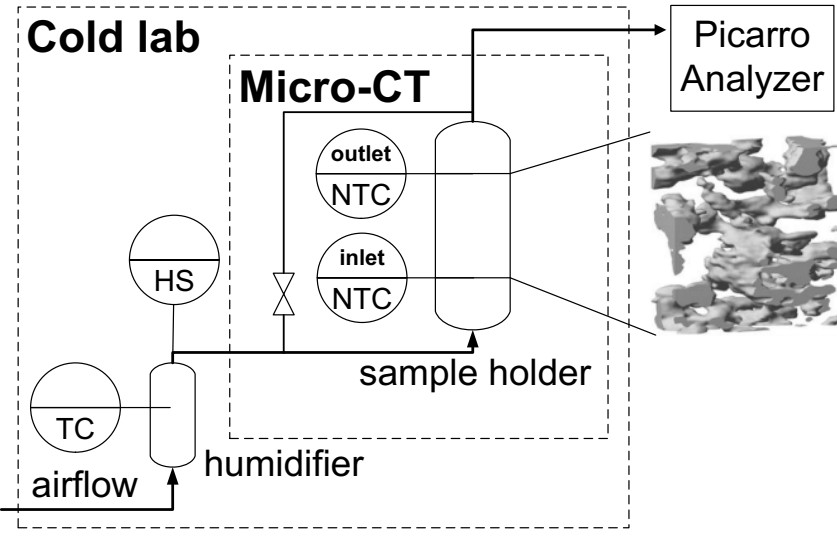


Fig. 1






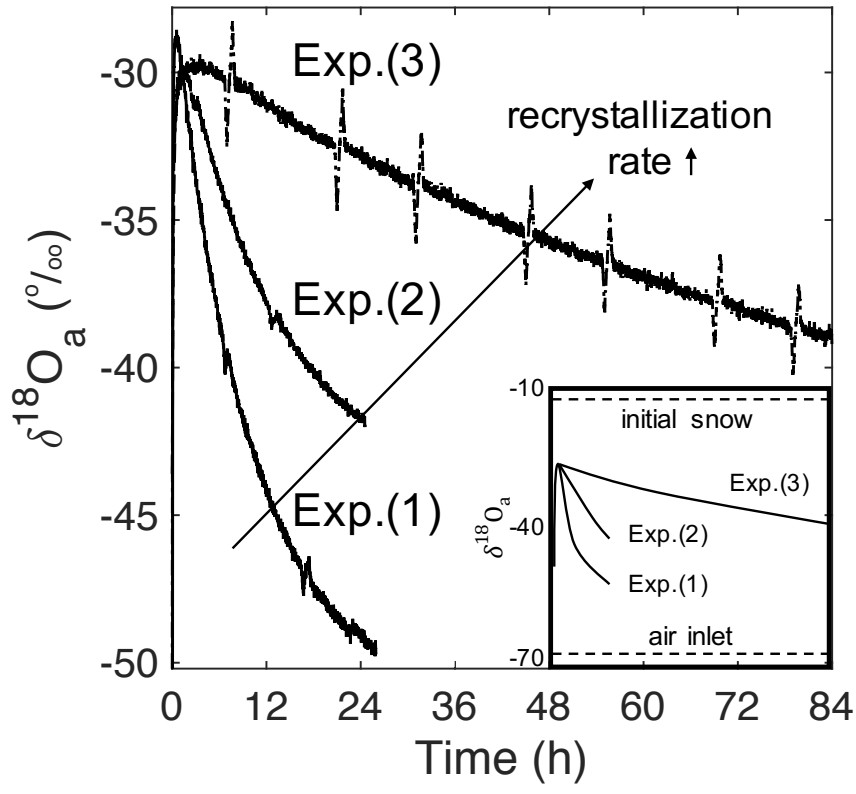


Fig. 2




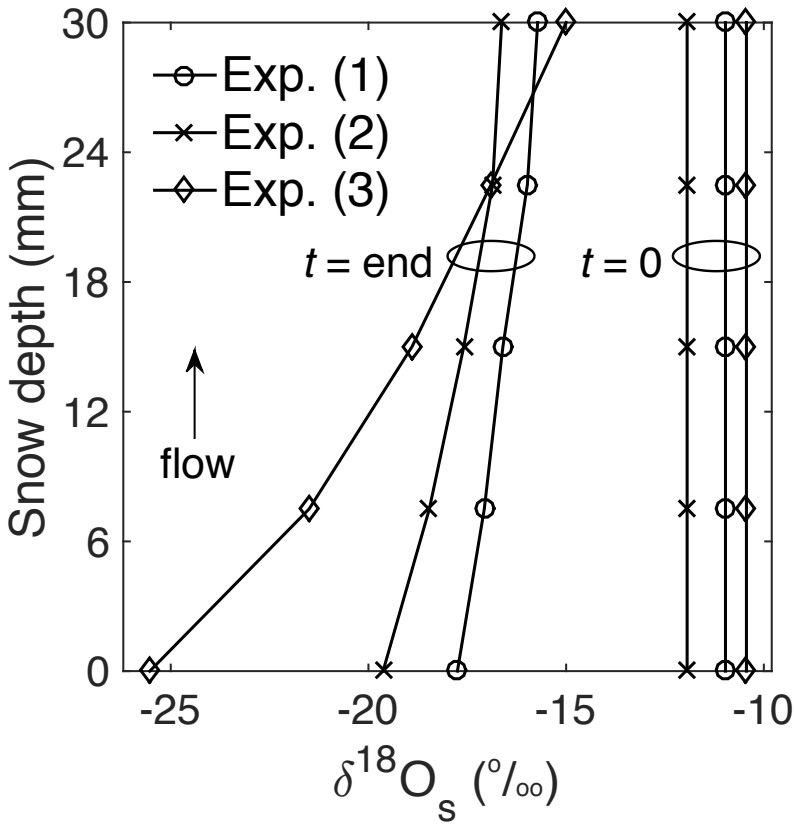


Fig. 3





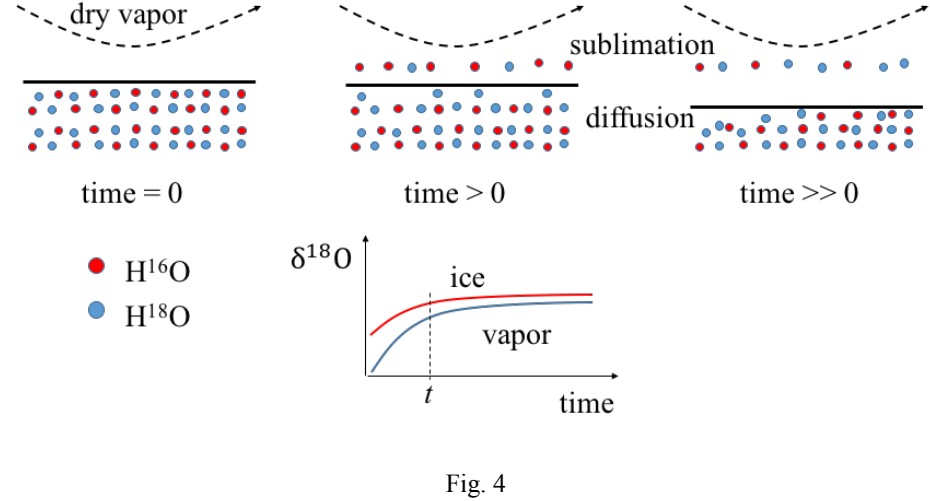


Fig. 4