# Peer review of "Experimental observation of transient $\delta^{18}\text{O}$ interaction between snow and advective airflow under various temperature gradient conditions"

_The Cryosphere, 2017_

## Referee Comment (RC1) · Anonymous Referee #1 · 4 Apr 2017

General comments: The submitted manuscript presents a really interesting laboratory investigation of the relationship between airflow, metamorphic processes, and the concentration of water stable isotopes in snow, with implication for the well-known proxy for temperature in ice core records of past climate conditions. Differing snow metamorphic properties were shown to impact the isotopic composition of snow with air being forced through it. This is a significant finding, and having the carefully-controlled laboratory experiments to support recent field observations is an important step in better understanding the controls of climate conditions on isotopic concentrations. Some of the observations made, namely that of sublimation and deposition either into the air flow-

ing through the sample or from the air onto the sample depending on the experimental conditions are really insightful and help to further explain these processes.

Specific comments: I have three comments that might require revision. The first comment is that it does seem like the Re number is high if it is 3 times that of natural conditions. It there a windspeed in natural conditions that this Re number relates to? The second comment is that some of the discussion and results sections seem more suited for the methodology or introduction sections (specific instances are detailed in the notes below, along with the technical comments I have). The third comment is that the reported density of the 3rd experiment is higher than the density of the samples in experiments 1 and 2 (which are oddly identical to 2 digits? were these taken from the same larger block of snow?). Given that the density is 10% higher for experiment 3 than the other two experiments, is there any possibility that this impacted the results at all?

Technical comments: There are a quite few mostly minor technical corrections to make to improve grammar, listed below by line number. Some are merely suggestions to improve readability.

Line 18: Suggest rewriting as "..study on the effect of airflow on snow isotopic composition"

Line 21: There should be a "the" in front of the phrase "exchange of isotopes"

Line 26: A highly resolved history of what is relevant? Not sure what authors are referring to here, think that it is a highly resolved climate history, maybe? Temperature and barometric conditions?

Line 35: Should be "the isotopic composition of high-latitude precipitation" or "the isotopic composition of precipitation in high-latitudes" not quite sure which one the authors meant.

Line 40: Suggest rewriting as "starting from the process of evaporation in the source

region, transportation of the air mass to the top of the ice sheet, and post-depositional processes..." only to make these phrases parallel.

Line 48: Suggest (very minor suggestion), replacing "nearby" with something more descriptive such as "closely spaced" or "closely located"

Line 56: Should be a comma after et al. in the citation

Line 63: don't need the dash in the phrase in between

Line 79-88: Another suggestion, but the beginning of this paragraph might make more sense further up in the introduction, before the paragraph discussing ice cores, i.e. before the paragraph starting on line 54. It seems out of place here, as the authors have just finished discussing self-diffusion in the ice matrix.

Line 85: What are the conditions for the typical half-life of a few days? I.e., normal mid-latitude conditions, or polar conditions?

Line 94: "small scale" should be hyphenated, "small-scale"

Line 95: have, should be "has", i.e. "Modeling....has assumed..." (or the authors could change it to "Models...have assumed..."

Line 107: Should be, "Here, we continuously measured..."

Line 122: Should be, "To prevent air flow between..." (delete extra "the")

Line 129: Delete "were" in front of the word "connected", i.e. should read "The experimental setup consisted of three main components....connected with insulated copper tubing..."

Line 138: Seems there is an extra word here, should be, "to limit the influence of variability" or "to limit changes in absolute vapor pressure"

Line 148: Suggest (minor suggestion) deleting the phrase "because it already had the appliance" mostly because this phrase is a little awkward, but also not really needed.

I'm not sure what the authors mean by "already had the appliance" unless they mean that it had already been applied for use in the micro-CT.

Line 163: The authors should define "opening size distribution" or describe how this is done.

Line 168: This phrase is used a few times in the manuscript, and it is slightly confusing, so I suggest rephrasing here and throughout (tried to make note of each instance), i.e. instead of "where cold air is heated up while flowing through the sample" maybe something along the lines of "where cold air entering the sample is heated while flowing..." For some reason, the phrase "heated up" is confusing. The same suggestion applies for next sentence, Line 170. I would suggest rewriting "where warm air entering the sample is cooled while flowing across the sample"

Line 177: "was" should be "were" since there are multiple temperatures being measured.

Line 185: Is the 0.7 % amount an average for the 3 experiments? It looks like it just pertains to Experiment 1, and the other experiments had lower increases, so maybe this could say, "A slight increase with a maximum of 0.7% of d18O in the water vapor produced by the humidifier was observed in experiment (1), with lower increases during experiments (2) and (3).

Line 193: the word "was" should be deleted (since "decreased" is the verb in the sentence).

Line 196: "memory effects" should be defined or explained. I think that the authors are referring to the having some water vapor in the lines that is either outside air, or the air that was left in the lines before the experiment started that needs to be purged. I also wonder if it is worth reporting these values, or if it could be better explained as part of the methodology, and not include the first 30 min of data in the plots.

Line 200: The authors use the phrase "was observed" twice in this sentence (and

throughout). I suggest changing the second "was observed" to something else, i.e. "as manifest by" so that it is not a repetitive

Line 212: This sentence uses "was observed" twice again. Suggest rewriting to something along the lines of, " As in the isothermal experiment (1), we observed a relaxing exponential decrease of d18Oa in the outlet flow through the measurement period..." (also, the way it is written is not quite grammatically correct, it should be, "Again, we observed..." but since that is a little vague, and not clear what "again" is referring to, I suggest elaborating a bit)

Line 217: again, the term "memory effects" should be defined at some point, probably at first mention of this idea.

Line 220: The phrase "decreased up to..." is a little confusing, suggest writing as "decrease in value from 4.66 to 7.66%..."

Line 221: There should be a "the" in front of the phrase "isotopic composition"

Line 239: As before, the phrase "decreased up to" is confusing. I think it could even just be "decreased 4.46 - 15.09%"

Line 251-257: This section of the paragraph seems out of place, like it belongs in the methods section. Also, it does seem like the Re number is high if it is 3 times that of natural conditions. It there a windspeed in natural conditions that this Re number relates to?

Line 262-268: This section of the paragraph seems like it would be better suited to the introduction than this section of the paper.

Line 273: This sentence is not grammatically correct as written, specifically, "will lead to influence the interpretation" is not correct. Suggest rewriting as, "will lead to improvement of the interpretation..."

Line 277: These citations should be written as "Persson et al., 2011 and Fujita and

Abe, 2006) to be consistent with the rest of the manuscript.

Line 290: I'm not sure what the authors mean by "changing the "target" toward which the snow is equilibrating." This sentence needs to be clarified and corrected.

Line 295: There is a missing phrase here, I think it should be something like, "Relatively short time exposed to vapor-snow exchange..." as it is written is a little confusing, and maybe grammatically incorrect (unless the first bit is meant as one long phrase).

Line 296: suggest adding the word "exposure" (or similar word) after "long -time" to clarify.

Line 305: This sentence is awkward, not necessarily incorrect. I suggest rewriting to "Despite a relatively small change in the difference..."

Line 311: suggest "is altered" instead of "gets altered"

Line 313: suggest adding the phrase "of the ice crystals" after "interior" just to clarify

Line 319: suggest changing "experiment" to "experiments" since there are 3 experiments discussed

Line 320: suggest instead of "an expected low altering" maybe "an expected minimal alteration" or "an expected minimal change"

Line 325: suggest deleting "(above)" and maybe replacing with, "...,as defined above,..."

Line 325: there is a missing "of" in front of the word "the"

Line 326: suggest rewriting these two sentences starting with "It is quite hard..." and ending with "the snow between experiment (1) and (2)" to something along the lines of, "There is a small, but notable, difference in the total d18O of the vapor..."

Line 338: suggest specifying the mechanism of redistribution of ice referred to, i.e. "this redistribution of ice caused by temperature gradient" (I think that is what the authors

are referring to, anyway.)

Line 359: instead of "there are complex interplays.." suggest changing to "causes a complex interplay"

Line 362: instead of "thus the change in the d18O of air" suggest, "and causes the change in the d18O of air"

Line 364: "there are deposition" should be "there is deposition"

Line 364: This observation of the sublimation and deposition either into the air flowing through the sample or from the air onto the sample depending on the experimental conditions is really cool.

Line 374: "in to" should be one word, "into"

Line 375-382: This is a really good discussion of the results.

Line 387: should have parentheses around "3" after "experiment" to be consistent

Line 390: suggest rewriting "By having this observation in mind," to "This observation, together..." since the phrase as written isn't grammatically correct, and is also colloquial.

Line 392: "lead" should be "leads" since it refers to "this observation," which is singular

Line 397: This sentence is not quite correct, and should be rewritten (since the hypotheses can't ask for anything). I suggest something like, "Our results and conclusions indicate that there is a need for additional validation..."

Line 399: This sentence is confusing since "scanning" defined. I think the authors mean that the entire sample should be scanned in the micro-CT, but then not sure what they mean by the phrase, "or gravimetrically." I'm confused because to determine the density of the snow gravimetrically would lead to a macro-scale measurement on the order of a few centimeters, when really, it seems like higher resolution measurements

are needed, i.e. grain-by-grain changes throughout the whole sample. I guess it would be good for the authors to define the scale at which the microstructure changes should be measured.

Line 402: "this would allow calculating more precisely the different observed exchange rates" should be "more precise calculation of the different observed exchange rates..."

Line 411: "helps" should be "help"

Line 412: I would suggest adding the phrase "under low accumulation conditions" after "better" to help tie into natural conditions that correspond to longer experimental times.

Line 412: this sentence is confusing, suggest rewriting as, " Further, because a complex interplay of sublimation and deposition was created within the humidifier as well as due to the geometrical complexity of snow..." somehow that isn't quite what was intended, but it is hard to understand what is meant by this sentence.

Line 415-418: This sentence is confusing, not really sure what to recommend. Suggest something like, "With the suggested experiment, whether or not there is measureable fractionation and if the sublimated vapor has the same d18O value as the sublimating ice can be determined."

Line 419: At the beginning of what? This experiment? Or also just after deposition?

Line 424: "an approach of the d18O value of the vapor and ice should be seen" is confusing, not sure what is meant by this.

Line 429: Suggest rewriting the sentence to, "This estimation suggests that after about 10ˆ3 second..."

Line 431: suggest deleting the last sentence, as the entire section was about laboratory experiments that are suggested based on the results of this work

Line 438-447: These are all details about the methodology that don't really belong in the summary. I think a lot of the details should be deleted, if not this whole paragraph,

and start the summary on line 448.

Line 455: think there is a missing word or phrase between "ice matrix" and "the temporal change", i.e. maybe should be, "the ice matrix causes the temporal change...at the outflow to decrease..."

Line 457: the same comment as above for this sentence, think that it should say, "Decreasing the recrystallization rate causes the temporal curve of the outlet concentration to become steeper,..."

Line 462: Is "cloud-temperature signal" a real term, or is this precipitation? If it is a real term, it should be defined. I am not familiar with that as a commonly used way to describe the temperature in the clouds.

Line 465-467 suggest deleting this first sentence in the paragraph...I think it is much stronger to say simply, "Our results represent the first direct experimental observation..." I would suggest replacing the word "showing" with the phrase "of the" If the authors do want to leave the first sentence, it is not grammatically correct, and should be, "These are novel measurements and will therefore be important as the basis for further research and experiments."

Line 470-471: Either this sentence should be broken into two sentences after the work "process," or I would suggest deleting the first part of the sentence, and simply saying, "Our results demonstrate that recrystallization and bulk mass exchange must be incorporated into future models..." That sentence is much stronger without the first phrase, as it avoids the repetition of the word "recrystallization."

Line 734 in Fig 2 caption: "run" should be "runs"

Fig 2: I am not sure what the large, long arrow is supposed to represent that is going diagonally across the figure. I think the little arrow is the "recrystallization rate" Should be explained, or labeled. Or is the long arrow recrystallization rate? Regardless, it is confusing.

Figure 3: What are the oblongs circling the lines supposed to represent? Are they supposed to be circling the "t=end" results vs. the "t=0" results to group them together somehow? I would recommend maybe different colors or something else. The oblongs are just confusing, and there is enough separation between the 2 groups of results that there must be another way to represent the time difference.

---

## Referee Comment (RC2) · Anonymous Referee #2 · 20 Apr 2017

The manuscript is devoted to the results of the laboratory experiments aimed to study the post-depositional changes of snow isotopic composition due to interaction of snow matrix with water vapor. The processes occurring in snow after the snow precipitation is deposited are one of the least studied and understood elements of the formation of the climatic signal of an ice core isotopic profile. Thus the present work is timely and up-to-date. The obtained results are clear and convincing so I think the manuscript may be accepted with minor corrections. My suggestions to improve the manuscript are as follows: I do not agree that the "results represent the first direct experimental observation showing interaction between the water isotopic composition of the snow" (line

467-468), since several similar laboratory experiments have been already contacted (e.g., Sokratov & Golubev, 2009). I think this work would benefit from short discussion of the previous studies. Second, I suggest to shorten or completely eliminate the long discussion of an experiment that has not been conducted yet (lines 412-432). Finally, some sentences look awkward or not finished. One of the examples is on the lines 361-362, but there are some more in the text. So I ask authors to look through the text more carefully.

Reference: Sokratov, S.A. and V.N. Golubev 2009. Snow isotopic content change by sublimation. J. Glaciol., 55(193): 823-828.

---

## Author Comment (AC1) · 15 May 2017

**RESPONSE TO ANONYMOUS REFEREE #1 COMMENTS**

**TO MANUSCRIPT tc-2017-16-RC1**

*Title:*     Experimental observation of transient δ18O interaction between snow and advective airflow under various temperature gradient conditions

**Authors:**  Pirmin Philipp Ebner, Hans Christian Steen-Larsen, Martin Schneebeli, Barbara Stenni, and Aldo Steinfeld

We thank the anonymous referee #1 for his constructive comments and suggestions. All line numbers correspond to the discussion paper.

**ANONYMOUS REVIEWER #1**

General comments: The submitted manuscript presents a really interesting laboratory investigation of the relationship between airflow, metamorphic processes, and the concentration of water stable isotopes in snow, with implication for the well-known proxy for temperature in ice core records of past climate conditions. Differing snow metamorphic properties were shown to impact the isotopic composition of snow with air being forced through it. This is a significant finding, and having the carefully-controlled laboratory experiments to support recent field observations is an important step in better understanding the controls of climate conditions on isotopic concentrations. Some of the observations made, namely that of sublimation and deposition either into the air flowing through the sample or from the air onto the sample depending on the experimental conditions are really insightful and help to further explain these processes.

Specific comments: I have three comments that might require revision. The first comment is that it does seem like the Re number is high if it is 3 times that of natural conditions. It there a windspeed in natural conditions that this Re number relates to? The second comment is that some of the discussion and results sections seem more suited for the methodology or introduction sections (specific instances are detailed in the notes below, along with the technical comments I have). The third comment is that the reported density of the 3rd experiment is higher than the density of the samples in experiments 1 and 2 (which are oddly identical to 2 digits? were these taken from the same larger block of snow?). Given that the density is 10% higher for experiment 3 than the other two experiments, is there any possibility that this impacted the results at all?

**Respond to main comment #1:** The Reynolds number in the experiment is not three times that of natural conditions but the airflow velocity inside the snow sample. Therefore, looking at the

Reynolds number our experiments are in the feasible flow regime (laminar flow) in a snow pack. The air velocity inside the snow sample of each experiment simulated high wind speed conditions above the snow surface (> 10 m s$^{-1}$).

**Respond to main comment #2:** We rearranged the different section and put some of the discussion and results sections in the methodology or introduction sections.

**Respond to main comment #3:** All the snow sample were taken from the same snow block, however it is quite hard to have the same identical density or porosity, respectively. Proksch et al (2016) report that the density in snow can only be determined with about 5% uncertainty. A 10% change in density affects the permeability (Zermatten et al., 2014) at the high porosity used here not in a non-linear way, i.e. not more than the 10% change. The effect of different porosities over a broad range would cleary have an impact on the results, but this has to be verified in further experiments, as mentioned at the end of the discussion section.

Technical comments: There are a quite few mostly minor technical corrections to make to improve grammar, listed below by line number. Some are merely suggestions to improve readability.

**Comment #1:** Line 18: Suggest rewriting as "..study on the effect of airflow on snow isotopic composition"

**Revision:** Text changed in the revised manuscript.

**Comment #2:** Line 21: There should be a "the" in front of the phrase "exchange of isotopes"

**Revision:** Text changed in the revised manuscript.

**Comment #3:** Line 26: A highly resolved history of what is relevant? Not sure what authors are referring to here, think that it is a highly resolved climate history, maybe? Temperature and barometric conditions?

**Revision:** The highly resolved climate history is relevant. Text changed in the revised manuscript.

Line 26: "… a highly resolved climate history …".

**Comment #4:** Line 35: Should be "the isotopic composition of high-latitude precipitation" or "the isotopic composition of precipitation in high-latitudes" not quite sure which one the authors meant.

**Revision:** Text changed in the revised manuscript.

Line 35: "… the isotopic composition of precipitation in high-latitude …".

**Comment #5:** Line 40: Suggest rewriting as "starting from the process of evaporation in the source region, transportation of the air mass to the top of the ice sheet, and post-depositional processes..." only to make these phrases parallel.

**Revision:** Text changed in the revised manuscript.

**Comment #6:** Line 48: Suggest (very minor suggestion), replacing "nearby" with something more descriptive such as "closely spaced" or "closely located"

**Revision:** "nearby" replaced with "closely located".

**Comment #7:** Line 56: Should be a comma after et al. in the citation

**Revision:** Comma added in the revised manuscript.

**Comment #8:** Line 63: don't need the dash in the phrase in between

**Revision:** Dash deleted in the revised manuscript.

**Comment #9:** Line 79-88: Another suggestion, but the beginning of this paragraph might make more sense further up in the introduction, before the paragraph discussing ice cores, i.e. before the paragraph starting on line 54. It seems out of place here, as the authors have just finished discussing self-diffusion in the ice matrix.

**Revision:** Text from line 79-88 moved to before the paragraph starting on line 54 in the revised manuscript.

**Comment #10:** Line 85: What are the conditions for the typical half-life of a few days? I.e., normal mid-latitude conditions, or polar conditions?

**Comment:** The half-life of a few days is defined by the temperature gradient. This condition can occur under mid-latitude or polar conditions. The conditions prevail in shallow snowpacks, as typical for snow on sea ice, and shallow tundra and taiga snow, as well as continental alpine snow.

**Revision:** Text changed in the revised manuscript.

> Line 86: "… is dictated by the temperature gradient and this can occur under mid-latitude conditions or polar conditions.".

**Comment #11:** Line 94: "small scale" should be hyphenated, "small-scale"

**Revision:** Hyphenate added in the revised manuscript.

**Comment #12:** Line 95: have, should be "has", i.e. "Modeling....has assumed..." (or the authors could change it to "Models...have assumed..."

**Revision:** Text changed in the revised manuscript.

> Line 95: "Models … have assumed …".

**Comment #13:** Line 107: Should be, "Here, we continuously measured..."

**Revision:** Text changed in the revised manuscript.

**Comment #14:** Line 122: Should be, "To prevent air flow between..." (delete extra "the")

**Revision:** Text changed in the revised manuscript.

**Comment #15:** Line 129: Delete "were" in front of the word "connected", i.e. should read "The experimental setup consisted of three main components....connected with insulated copper tubing..."

**Revision:** Text changed in the revised manuscript.

**Comment #16:** Line 138: Seems there is an extra word here, should be, "to limit the influence of variability" or "to limit changes in absolute vapor pressure"

**Revision:** Text changed in the revised manuscript.

> Line 138: "… to limit the influence of variability …".

**Comment #17:** Line 148: Suggest (minor suggestion) deleting the phrase "because it already had the appliance" mostly because this phrase is a little awkward, but also not really needed. I'm not sure what the authors mean by "already had the appliance" unless they mean that it had already been applied for use in the micro-CT.

**Revision:** Phrase deleted in the revised manuscript.

**Comment #18:** Line 163: The authors should define "opening size distribution" or describe how this is done.

**Revision:** Text added in the revised manuscript.

> Line 163: "The opening size distribution can be imagined as virtual sieving with different mesh size.".

**Comment #19:** Line 168: This phrase is used a few times in the manuscript, and it is slightly confusing, so I suggest rephrasing here and throughout (tried to make note of each instance), i.e. instead of "where cold air is heated up while flowing through the sample" maybe something along the lines of "where cold air entering the sample is heated while flowing..." For some reason, the phrase "heated up" is confusing. The same suggestion applies for next sentence, Line

170. I would suggest rewriting "where warm air entering the sample is cooled while flowing across the sample"

**Revision:** Text changed in the revised manuscript.

**Comment #20:** Line 177: "was" should be "were" since there are multiple temperatures being measured.

**Revision:** Text changed in the revised manuscript.

**Comment #21:** Line 185: Is the 0.7 % amount an average for the 3 experiments? It looks like it just pertains to Experiment 1, and the other experiments had lower increases, so maybe this could say, "A slight increase with a maximum of 0.7% of d18O in the water vapor produced by the humidifier was observed in experiment (1), with lower increases during experiments (2) and (3).

**Revision:** Text changed in the revised manuscript.

**Comment #22:** Line 193: the word "was" should be deleted (since "decreased" is the verb in the sentence).

**Revision:** Text changed in the revised manuscript.

**Comment #23:** Line 196: "memory effects" should be defined or explained. I think that the authors are referring to the having some water vapor in the lines that is either outside air, or the air that was left in the lines before the experiment started that needs to be purged. I also wonder if it is worth reporting these values, or if it could be better explained as part of the methodology, and not include the first 30 min of data in the plots.

**Revision:** We didn't delete the "memory effects" in the result because this term is quite commonly used in the literature (e.g. Penna et al., 2012) but we moved the part into the methodology section.

Penna, D., Stenni, B., Šanda, M., Wrede, S., Bogaard, T. A., Michelini, M., et al. (2012). Technical Note: Evaluation of between-sample memory effects in the analysis of δ2H and δ18O of water samples measured by laser spectroscopes. Hydrology and Earth System Sciences, 16(10), 3925–3933. http://doi.org/10.5194/hess-16-3925-2012.

> Line 189: "In the first approximately 30 min, the isotopic composition of the measured outflow air $\delta^{18}O_a$ increased from a low $\delta^{18}O$ to a starting value of around -29‰ in each experiment. This was due to memory effect possible condensed water left in the tubes from a prior experiment.".

**Comment #24:** Line 200: The authors use the phrase "was observed" twice in this sentence (and throughout). I suggest changing the second "was observed" to something else, i.e. "as manifest by" so that it is not a repetitive

**Revision:** Text changed in the revised manuscript.

> Line 200: "… as manifest by …".

**Comment #25:** Line 212: This sentence uses "was observed" twice again. Suggest rewriting to something along the lines of, " As in the isothermal experiment (1), we observed a relaxing exponential decrease of d18Oa in the outlet flow through the measurement period..." (also, the way it is written is not quite grammatically correct, it should be, "Again, we observed..." but since that is a little vague, and not clear what "again" is referring to, I suggest elaborating a bit)

**Revision:** Text changed in the revised manuscript.

> Line 212: "As in the isothermal experiment (1), we observed …".

**Comment #26:** Line 217: again, the term "memory effects" should be defined at some point, probably at first mention of this idea.

**Revision:** See comment #23.

**Comment #27:** Line 220: The phrase "decreased up to..." is a little confusing, suggest writing as "decrease in value from 4.66 to 7.66%..."

**Revision:** Text changed in the revised manuscript.

**Comment #28:** Line 221: There should be a "the" in front of the phrase "isotopic composition"

**Revision:** Text changed in the revised manuscript.

**Comment #29:** Line 239: As before, the phrase "decreased up to" is confusing. I think it could even just be "decreased 4.46 - 15.09%"

**Revision:** Text changed in the revised manuscript.

**Comment #30:** Line 251-257: This section of the paragraph seems out of place, like it belongs in the methods section. Also, it does seem like the Re number is high if it is 3 times that of natural conditions. It there a windspeed in natural conditions that this Re number relates to?

**Comment:** See responds to main comment #1.

**Revision:** Text moved to the method section (line 175) and text additionally added in the revised manuscript.

> Line 175: "In wind pumping theory, an airflow velocity of $u_d \approx 10$ mm s$^{-1}$ (corresponding

Reynolds number Re ≈ 0.65) was estimated inside the surface snow layers ($d_{mean}$ ≈ 1 mm) for a high wind speed above the snow surface (≈ 10 m s$^{-1}$) (Neumann, 2003). We performed experiments with airflow velocities inside the snow sample of around 30 mm s$^{-1}$ (corresponding Reynolds number Re = 0.7), which was a factor three higher than in the wind pumping theory. But, looking at the Reynolds number our experiments were in the feasible flow regime (laminar flow) of a natural snow pack.".

**Comment #31:** Line 262-268: This section of the paragraph seems like it would be better suited to the introduction than this section of the paper.

**Revision:** This section of the paragraph is moved to the introduction paragraph (line 91) in the revised manuscript.

**Comment #32:** Line 273: This sentence is not grammatically correct as written, specifically, "will lead to influence the interpretation" is not correct. Suggest rewriting as, "will lead to improvement of the interpretation..."

**Revision:** Text changed in the revised manuscript.

**Comment #33:** Line 277: These citations should be written as "Persson et al., 2011 and Fujita and Abe, 2006) to be consistent with the rest of the manuscript.

**Revision:** Citation changed in the revised manuscript.

**Comment #34:** Line 290: I'm not sure what the authors mean by "changing the "target" toward which the snow is equilibrating." This sentence needs to be clarified and corrected.

**Comment:** It's a bit confusing, it describe the same principle as mention before. We deleted this in the revised manuscript

**Comment #35:** Line 295: There is a missing phrase here, I think it should be something like, "Relatively short time exposed to vapor-snow exchange..." as it is written is a little confusing, and maybe grammatically incorrect (unless the first bit is meant as one long phrase).

**Revision:** Text changed in the revised manuscript.

Line 125: "Relatively short time exposed to vapor …".

**Comment #36:** Line 296: suggest adding the word "exposure" (or similar word) after "long -time" to clarify.

**Revision:** Word  added in the revised manuscript.

**Comment #37:** Line 305: This sentence is awkward, not necessarily incorrect. I suggest rewriting to "Despite a relatively small change in the difference..."

**Revision:** Text changed in the revised manuscript.

**Comment #38:** Line 311: suggest "is altered" instead of "gets altered"

**Revision:** Text changed in the revised manuscript.

**Comment #39:** Line 313: suggest adding the phrase "of the ice crystals" after "interior" just to clarify

**Revision:** Text changed in the revised manuscript.

**Comment #40:** Line 319: suggest changing "experiment" to "experiments" since there are 3 experiments discussed

**Revision:** Word changed in the revised manuscript.

**Comment #41:** Line 320: suggest instead of "an expected low altering" maybe "an expected minimal alteration" or "an expected minimal change"

**Revision:** Text changed in the revised manuscript.

**Comment #42:** Line 325: suggest deleting "(above)" and maybe replacing with, "...,as defined above,..."

**Revision:** Text changed in the revised manuscript.

**Comment #43:** Line 325: there is a missing "of" in front of the word "the"

**Revision:** Text changed in the revised manuscript.

**Comment #44:** Line 326: suggest rewriting these two sentences starting with "It is quite hard..." and ending with "the snow between experiment (1) and (2)" to something along the lines of, "There is a small, but notable, difference in the total d18O of the vapor..."

**Revision:** Text changed in the revised manuscript.

**Comment #45:** Line 338: suggest specifying the mechanism of redistribution of ice referred to, i.e. "this redistribution of ice caused by temperature gradient" (I think that is what the authors are referring to, anyway.)

**Revision:** Text changed in the revised manuscript.

**Comment #46:** Line 359: instead of "there are complex interplays.." suggest changing to "causes a complex interplay"

**Revision:** Text changed in the revised manuscript.

**Comment #47:** Line 362: instead of "thus the change in the d18O of air" suggest, "and causes the change in the d18O of air"

**Revision:** Text changed in the revised manuscript.

**Comment #48:** Line 364: "there are deposition" should be "there is deposition"

**Revision:** Text changed in the revised manuscript.

**Comment #49:** Line 364: This observation of the sublimation and deposition either into the air flowing through the sample or from the air onto the sample depending on the experimental conditions is really cool.

**Comment:** Thanks, further information can be found in the paper Ebner et al., 2015b and Ebner et al., 2015c.

**Comment #50:** Line 374: "in to" should be one word, "into"

**Revision:** Word changed in the revised manuscript.

**Comment #51:** Line 375-382: This is a really good discussion of the results.

**Comment:** Thanks.

**Comment #52:** Line 387: should have parentheses around "3" after "experiment" to be consistent

**Revision:** Text changed in the revised manuscript.

**Comment #53:** Line 390: suggest rewriting "By having this observation in mind," to "This observation, together..." since the phrase as written isn't grammatically correct, and is also colloquial.

**Revision:** Text changed in the revised manuscript.

**Comment #54:** Line 392: "lead" should be "leads" since it refers to "this observation," which is singular

**Revision:** Text changed in the revised manuscript.

**Comment #55:** Line 397: This sentence is not quite correct, and should be rewritten (since the hypotheses can't ask for anything). I suggest something like, "Our results and conclusions indicate that there is a need for additional validation..."

**Revision:** Text changed in the revised manuscript.

**Comment #56:** Line 399: This sentence is confusing since "scanning" defined. I think the authors mean that the entire sample should be scanned in the micro-CT, but then not sure what they mean by the phrase, "or gravimetrically." I'm confused because to determine the density of the snow gravimetrically would lead to a macro-scale measurement on the order of a few centimeters, when really, it seems like higher resolution measurements are needed, i.e. grain-by-grain changes throughout the whole sample. I guess it would be good for the authors to define the scale at which the microstructure changes should be measured.

**Revision:** Text added in the revised manuscript.

Line 399: "Ideally, the entire sample would be tomographically measured with a resolution of 4 x 4 x 4 mm$^3$, each cube corresponding to the representative volume.".

**Comment #57:** Line 402: "this would allow calculating more precisely the different observed exchange rates" should be "more precise calculation of the different observed exchange rates..."

**Revision:** Text changed in the revised manuscript.

**Comment #58:** Line 411: "helps" should be "help"

**Revision:** Text changed in the revised manuscript.

**Comment #59:** Line 412: I would suggest adding the phrase "under low accumulation conditions" after "better" to help tie into natural conditions that correspond to longer experimental times.

**Revision:** Text changed in the revised manuscript.

**Comment #60:** Line 412: this sentence is confusing, suggest rewriting as, " Further, because a complex interplay of sublimation and deposition was created within the humidifier as well as due to the geometrical complexity of snow..." somehow that isn't quite what was intended, but it is hard to understand what is meant by this sentence.

**Revision:** We removed the line 412-432 and Figure 4, as it takes up not completely resolved questions already posed in the paper by Horita et al. (2008) (Line 412-432).

Horita J., Rozanski K., and Cohen S.: Isotope effects in the evaporation of water: a status report of the Craig-Gordon model, Isot. Environ. Health Stud., 44, 23-49, 2008.

**Comment #61:** Line 415-418: This sentence is confusing, not really sure what to recommend. Suggest something like, "With the suggested experiment, whether or not there is measureable fractionation and if the sublimated vapor has the same d18O value as the sublimating ice can be determined."

**Revision:** See comment #60.

**Comment #62:** Line 419: At the beginning of what? This experiment? Or also just after deposition?

**Revision:** See comment #60.

**Comment #63:** Line 424: "an approach of the d18O value of the vapor and ice should be seen" is confusing, not sure what is meant by this.

**Revision:** See comment #60.

**Comment #64:** Line 429: Suggest rewriting the sentence to, "This estimation suggests that after about 10ˆ3 second..."

**Revision:** See comment #60.

**Comment #65:** Line 431: suggest deleting the last sentence, as the entire section was about laboratory experiments that are suggested based on the results of this work

**Revision:** See comment #60.

**Comment #66:** Line 438-447: These are all details about the methodology that don't really belong in the summary. I think a lot of the details should be deleted, if not this whole paragraph, and start the summary on line 448.

**Revision:** Paragraph deleted in the revised manuscript.

**Comment #67:** Line 455: think there is a missing word or phrase between "ice matrix" and "the temporal change", i.e. maybe should be, "the ice matrix causes the temporal change...at the outflow to decrease..."

**Revision:** Text changed in the revised manuscript.

**Comment #68:** Line 457: the same comment as above for this sentence, think that it should say, "Decreasing the recrystallization rate causes the temporal curve of the outlet concentration to become steeper, ..."

**Revision:** Text changed in the revised manuscript.

**Comment #69:** Line 462: Is "cloud-temperature signal" a real term, or is this precipitation? If it is a real term, it should be defined. I am not familiar with that as a commonly used way to describe the temperature in the clouds.

**Revision:** Text changed in the revised manuscript.

Line 462: "… can be superimposed on precipitation signal".

**Comment #70:** Line 465-467 suggest deleting this first sentence in the paragraph...I think it is much stronger to say simply, "Our results represent the first direct experimental observation..." I would suggest replacing the word "showing" with the phrase "of the" If the authors do want to leave the first sentence, it is not grammatically correct, and should be, "These are novel measurements and will therefore be important as the basis for further research and experiments."

**Revision:** We want to leave the first sentence. Text changed in the revised manuscript.

> Line 465-467: "These are novel measurements and will therefore be important as the basis for further research and experiments. Our results represent the first direct experimental observation of the interaction …".

**Comment #71:** Line 470-471: Either this sentence should be broken into two sentences after the work "process," or I would suggest deleting the first part of the sentence, and simply saying, "Our results demonstrate that recrystallization and bulk mass exchange must be incorporated into future models..." That sentence is much stronger without the first phrase, as it avoids the repetition of the word "recrystallization."

**Revision:** We deleted the first part of the sentence and changed the text in the revised manuscript.

**Comment #72:** Line 734 in Fig 2 caption: "run" should be "runs"

**Revision:** Text changed in the revised manuscript.

**Comment #73:** Fig 2: I am not sure what the large, long arrow is supposed to represent that is going diagonally across the figure. I think the little arrow is the "recrystallization rate" Should be explained, or labeled. Or is the long arrow recrystallization rate? Regardless, it is confusing.

**Revision:** Figure 2 changed in the revised manuscript.

[Figure]

**Comment #74:** Figure 3: What are the oblongs circling the lines supposed to represent? Are they supposed to be circling the "t=end" results vs. the "t=0" results to group them together somehow? I would recommend maybe different colors or something else. The oblongs are just confusing, and there is enough separation between the 2 groups of results that there must be another way to represent the time difference.

**Revision:** Figure changed in the revised manuscript.

[Figure]

Minor revisions were made throughout the revised manuscript.

We thank Anonymous Referee #1 for his insight, suggestions and recommendations.

The authors

References

Proksch, M., N. Rutter, C. Fierz, and M. Schneebeli (2016), Intercomparison of snow density measurements: bias, precision and spatial resolution, *Cryosph.*, *10*, 371–384, doi:10.5194/tc-10-371-2016.

---

## Author Comment (AC2) · 15 May 2017

*Title:* Experimental observation of transient δ18O interaction between snow and advective airflow under various temperature gradient conditions

**Authors:** Pirmin Philipp Ebner, Hans Christian Steen-Larsen, Martin Schneebeli, Barbara Stenni, and Aldo Steinfeld

We thank the anonymous referee #2 for his very constructive comments and suggestions. All line numbers correspond to the discussion paper.

**ANONYMOUS REVIEWER #2**

The manuscript is devoted to the results of the laboratory experiments aimed to study the post-depositional changes of snow isotopic composition due to interaction of snow matrix with water vapor. The processes occurring in snow after the snow precipitation is deposited are one of the least studied and understood elements of the formation of the climatic signal of an ice core isotopic profile. Thus the present work is timely and up-to-date. The obtained results are clear and convincing so I think the manuscript may be accepted with minor corrections. My suggestions to improve the manuscript are as follows: I do not agree that the "results represent the first direct experimental observation showing interaction between the water isotopic composition of the snow" (line 467-468), since several similar laboratory experiments have been already contacted (e.g., Sokratov & Golubev, 2009). I think this work would benefit from short discussion of the previous studies. Second, I suggest to shorten or completely eliminate the long discussion of an experiment that has not been conducted yet (lines 412-432). Finally, some sentences look awkward or not finished. One of the examples is on the lines 361-362, but there are some more in the text. So I ask authors to look through the text more carefully.

Reference: Sokratov, S.A. and V.N. Golubev 2009. Snow isotopic content change by sublimation. J. Glaciol., 55(193): 823-828.

**Response to comment #1:** We added a short discussion of the previous studies.

Line 262: "The results also support the statement that an interplay between theoretically expected layer-by-layer sublimation and deposition at the ice-matrix surface and the isotopic content evolution of snow cover due to mass exchange between the snow cover and the atmosphere occurs (Sokratov and Golubev, 2009). The specific surface area of

snow exposed to mass exchange (Horita et al., 2008) and by the depth of the snow layer exposed to the mass exchange with the atmosphere (He and Smith, 1999) plays an important role.".

Citation added

Sokratov S.A. and Golubev V. N.: Snow isotopic content change by sublimation. Journal of Glaciology, 55(193), 823-828, 2009.

Horita J., Rozanski K., and Cohen S.: Isotope effects in the evaporation of water: a status report of the Craig-Gordon model, Isot. Environ. Health Stud., 44, 23-49, 2008.

He H. and Smith R. B.: An advective-diffusive isotopic evaporation-condensation model, Journal of Geophysical Research, 104, 18619-18630, 1999.

And we changed the sentence in line 467-468

Line 467-468: "Our results represent direct experimental observation …".

**Response to comment #2:** We agree with the reviewer and removed this description, as it takes up not completely resolved questions already posed in the paper by Horita et al. (2008) (Line 412-432).

**Response to comment #3:** Thank you for the comment. We will check and correct the manuscript where appropriate (see revised manuscript).

Minor revisions were made throughout the revised manuscript.

We thank Anonymous Referee #2 for his insight, suggestions and recommendations.

The authors

---

## Author Response (AR2)

**RESPONSE TO EDITOR COMMENTS**

**TO MANUSCRIPT tc-2017-16-RC1**

*Title:* Experimental observation of transient δ18O interaction between snow and advective airflow under various temperature gradient conditions

**Authors:** Pirmin Philipp Ebner, Hans Christian Steen-Larsen, Martin Schneebeli, Barbara Stenni, and Aldo Steinfeld

We thank the Editor for his constructive comments and suggestions. All line numbers correspond to the annotated manuscript.

**Comment** The authors have responded well to both referees comments, and have made extensive revisions to their manuscript. I have identified only a couple of places where I think the manuscript needs attention. These should be easy to resolve before final publication.

**Comment #1:** There needs to be an explanation in the text for the identical density and porosity between experiment 1 and experiment 2. Even if the two specimens come from the same bulk sample, it would be surprising if they had the exact same density-- was the density measured on the bulk sample but not on the experimental specimens? In any case, if the density can only be determined to 5%, far too many significant precisions are included in table 1.

**Revision:** All snow samples were taken from the same snow block but the estimated porosity and density in Table 1 are the exact measured snow sample in each experiment by micro-CT. We reduced the number of decimals in Table 1 to a more realistic precision. We added a text in the methods part and in the caption of table 1.

"All snow samples were taken from the same snow block with an average density of ≈ 210 kg m$^{-3}$. The estimated density given in Table 1 was the density of the snow sample in each experiment measured by μCT.".

"Table 1: Morphological properties and flow characteristics of the experimental runs: μCT measured snow density ($\rho$), porosity ($\varepsilon$), specific surface area per unit mass (SSA), mean pore space diameter ($d_{mean}$), …"

**Comment #2:** The wording of the revision in response to R1's comment 18 is a bit confusing: I would recommend writing: "The mean pore size distribution(d_mean) was estimated using the opening-size-distribution operation. This operation can be imagined..."

**Revision:** Text changed in the revised manuscript.

**Comment #3:** The revision in response to R1's comment 23 is not clear. I think there are some words missing from the second sentence, and the authors need to be clear whether the memory effect is known to have been important, or whether it is one possible effect that might have produced the observer result.

**Revision:** We changed this part and added a reference that the memory effect is known but not relevant for the results.

> "In the first approximately 30 min, the isotopic composition of the measured outflow air $\delta 18Oa$ increased from a low $\delta 18O$ to a starting value of around -29‰ in each experiment. This was due to memory effect and one possible effect might be condensed water left in the tubes from a prior experiment which had no further impact on the experiments (Penna et al., 2012).".

Reference added:

> Penna, D., Stenni, B., Šanda, M., Wrede, S., Bogaard, T. A., Michelini, M., et al. (2012). Technical Note: Evaluation of between-sample memory effects in the analysis of $\delta 2H$ and $\delta 18O$ of water samples measured by laser spectroscopes. Hydrology and Earth System Sciences, 16(10), 3925–3933. http://doi.org/10.5194/hess-16-3925-2012.

**Comment #4:** At line 196 of the annotated manuscript, there's a sentence that begins "In wind pumping theory..." I'd recommend rewriting this sentence in the active voice to make clear who used the wind-pumping theory, and then make clear whether your calculation of Re also uses the same wind-pumping theory.

**Revision:** Text changed in the revised manuscript.

> "We performed the experiments with airflow velocities in the snow sample at $u_D \approx 30$ mm s$^{-1}$, which is a factor of three higher than calculated by Neumann (2003) for a natural snow pack. But looking at the Reynolds number, describing the flow regime inside the pores, our experiments (Re $\approx 0.7$) were in the feasible flow regime (laminar flow) of a natural snow pack (Re $\approx 0.65$)."

Minor revisions were made throughout the revised manuscript.

We thank the Editor for his insight, suggestions and recommendations.

The authors

[revised manuscript text omitted]

                                   Fig. 1

[Figure]

                           Fig. 2

[Figure]

                                    Fig. 3